# Wilson and Jungner Revisited: Are Screening Criteria Fit for the 21st Century?

**DOI:** 10.3390/ijns10030062

**Published:** 2024-09-13

**Authors:** Elena Schnabel-Besson, Ulrike Mütze, Nicola Dikow, Friederike Hörster, Marina A. Morath, Karla Alex, Heiko Brennenstuhl, Sascha Settegast, Jürgen G. Okun, Christian P. Schaaf, Eva C. Winkler, Stefan Kölker

**Affiliations:** 1Division of Pediatric Neurology and Metabolic Medicine, Department of Pediatrics I, Medical Faculty of Heidelberg, Heidelberg University, 69120 Heidelberg, Germany; 2Institute of Human Genetics, University Hospital Heidelberg, Heidelberg University, 69120 Heidelberg, Germany; 3Section Translational Medical Ethics, Department of Medical Oncology, National Center for Tumor Diseases (NCT), Medical Faculty of Heidelberg, Heidelberg University, 69120 Heidelberg, Germany

**Keywords:** newborn screening, neonatal screening, genomic newborn screening, screening criteria, public health program, phenotypic diversity, newborn sequencing

## Abstract

Driven by technological innovations, newborn screening (NBS) panels have been expanded and the development of genomic NBS pilot programs is rapidly progressing. Decisions on disease selection for NBS are still based on the Wilson and Jungner (WJ) criteria published in 1968. Despite this uniform reference, interpretation of the WJ criteria and actual disease selection for NBS programs are highly variable. A systematic literature search [PubMED search “Wilson” AND “Jungner”; last search 16.07.22] was performed to evaluate the applicability of the WJ criteria for current and future NBS programs and the need for adaptation. By at least two reviewers, 105 publications (systematic literature search, N = 77; manual search, N = 28) were screened for relevant content and, finally, 38 publications were evaluated. Limited by the study design of qualitative text analysis, no statistical evaluation was performed, but a structured collection of reported aspects of criticism and proposed improvements was instead collated. This revealed a set of general limitations of the WJ criteria, such as imprecise terminology, lack of measurability and objectivity, missing pediatric focus, and absent guidance on program management. Furthermore, it unraveled specific aspects of criticism on clinical, diagnostic, therapeutic, and economical aspects. A major obstacle was found to be the incompletely understood natural history and phenotypic diversity of rare diseases prior to NBS implementation, resulting in uncertainty about case definition, risk stratification, and indications for treatment. This gap could be closed through the systematic collection and evaluation of real-world evidence on the quality, safety, and (cost-)effectiveness of NBS, as well as the long-term benefits experienced by screened individuals. An integrated NBS public health program that is designed to continuously learn would fulfil these requirements, and a multi-dimensional framework for future NBS programs integrating medical, ethical, legal, and societal perspectives is overdue.

## 1. Introduction

Newborn screening (NBS) is considered a highly successful population-based measure of secondary prevention, challenging the traditional paradigms of medicine. By identifying affected individuals during the pre-clinical or early stage of a disease, NBS redirects the initiation of treatment to an early, commonly symptom-free period of life [1].

NBS programs started more than 60 years ago, with phenylketonuria as the first target disease [2]. Since then, the continuous extension of NBS programs has been driven by technological innovations such as tandem mass spectrometry [3,4], which exemplarily led to the inclusion of 29 primary and 25 secondary conditions in the American Recommended Uniform Screening Panel (RUSP) in 2006 [5], as well as multiple-tier strategies and molecular genetic tests in recent years [6,7,8,9]. The long-term observation of extended NBS cohorts has highlighted health benefits for screened newborns with a target disease, but also opportunities for further improvement [10,11,12,13,14,15,16,17,18].

“All screening programs do harm; some do good as well, and, of these, some do more good than harm at reasonable cost” [19]. In a nutshell, this is the potentially most condensed description of an NBS program. Since about 99.9% of currently screened newborns are not affected by a target disease, it is important to minimize their risk of harm while aiming to maximize the benefits for the 0.1% with a target disease. This is particularly important since current technological innovations are expected to be ready in the near future; this will set off an avalanche of new diseases that could be identified by NBS, but the framework guiding the development of NBS is not up to date. As sequencing technologies are rapidly advancing, they are expected to pave the way for the next substantial extension of future NBS programs [20,21]. A genomic NBS (gNBS) program would technically allow the early identification of hundreds of additional genetic target diseases that cannot be screened with currently applied NBS methods, as exemplified by current pilot gNBS studies including a median of 480 gene–disease combinations [22].

In addition to the potential benefits, gNBS will also come with the risk of harm [19]. The high acceptability of current NBS programs would be compromised if a multi-dimensional framework for gNBS was not carefully set and the selection of new target diseases was not based on a set of transparent NBS criteria agreed by the general public, while also addressing ethical, legal, and societal aspects [23]. As some countries have already initiated gNBS pilots [20,24,25,26,27,28,29,30,31], there is an urgent need to make the criteria for selecting NBS target diseases fit for this upcoming challenge.

So far, the decision on disease selection for NBS has generally been based on the screening principles of Wilson and Jungner (WJ) published in 1968 [32], i.e., at a time when NBS was still in its infancy. Despite applying the same set of criteria, national NBS programs differ greatly in their disease panels, highlighting that the WJ criteria are incomplete and leave a margin for interpretation. As a consequence, their capability to appropriately guide the development of NBS programs has been doubted repeatedly and a revision seems overdue [33,34,35,36,37]; however, various attempts to update the WJ criteria have not had a lasting effect on the development of NBS programs [33,34,35,36,37].

Facing a possible future gNBS, the present work aims to evaluate the current literature via a systematic review, focusing on the applicability of the original WJ criteria in the current NBS process, proposed modifications to existing criteria, and proposed additional criteria and sub-categories.

## 2. Materials and Methods

A systematic literature search following the PRISMA 2020 guidelines [38,39] was conducted (Prisma 2020 flow; Figure 1). The PRISMA 2020 checklist was used to prepare the manuscript. A literature search on articles published in PubMed between 1st January 2002 and 17th June 2022 with search criteria (“Wilson” [Title/Abstract]” AND “Jungner” [Title/Abstract]; last search 17.06.2022) was conducted, complemented by a manual search of selected scientific journals and books (Figure 1) and a review of the references cited in the selected literature. Prior to inclusion in the qualitative analysis, the abstracts of all identified publications were screened by at least two reviewers independently to identify whether the publications addressed at least one of the following aspects: (1) applicability of the original WJ criteria in the NBS process, (2) proposed modifications to existing criteria, (3) proposed additional screening criteria, or (4) domains of screening criteria.

In analogy to previous studies [33,36,37,40], we clustered the WJ criteria into four sub-categories, i.e., (I) clinical aspects, (II) diagnostic aspects, (III) therapeutic aspects, and (IV) economical aspects. For the qualitative analysis, again by at least two reviewers independently, results (citations and summaries) were sorted according to the following major topics: (A) general limitations of the original screening criteria, (B) criticism specifically addressing one of the four sub-categories or a single criterion, and (C) proposals for the inclusion of missing criteria and sub-categories. A table of all results and the groupings was reviewed and discussed by the group of reviewers together (N = 6), and all ambiguities were re-checked in the literature.

## 3. Results

Seventy-seven publications from the systematic literature search and twenty-eight publications from the manual search (Figure 1) were included in this study. The abstracts of these 105 publications were screened for relevant content with a focus on gNBS. Finally, 38 reports were included in the qualitative analysis (Figure 1) [5,33,34,35,36,37,40,41,42,43,44,45,46,47,48,49,50,51,52,53,54,55,56,57,58,59,60,61,62,63,64,65,66,67,68,69,70,71].

### 3.1. General Limitations of the WJ Criteria

The WJ criteria were developed with focus on adult screening programs and did not give explicit attention to the NBS emerging at that time. As a consequence, several studies highlight general limitations of the WJ criteria, such as that they “largely ignore the family dimensions inherent in NBS” [57]. Furthermore, the criteria and goals of NBS are not described in a specific, measurable, achievable, reasonable, and time-bound (SMART) way that allows effective goal-setting, objective development, and performance review [72]. This is reflected by the use of imprecise, unspecific, and “largely subjective” terminology [65], as well as a lack of clear quantification and guidance, suggesting that the WJ criteria were not intended to serve as a checklist to guide the selection of target diseases for NBS programs. Since then, several attempts have been made to update the screening criteria [33,34,35,36,37]; however, these studies rarely include a satisfactory solution for a transparent, quantifiable, and objective selection process for target diseases using semi-quantitative metrics or scoring systems [37,73].

### 3.2. Specific Aspects of Criticism of WJ Criteria

#### 3.2.1. Clinical Aspects

The clinical aspects addressed by the WJ criteria relate to natural history and disease course (criteria #1, #4, #7; Figure 2). With the knowledge from current NBS programs, some studies highlight that it is an impossible task to sufficiently describe the natural history and phenotypic diversity of a rare disease prior to NBS due to an inherent lack of data. Therefore, “any prior understanding of disease is inevitably found to be insufficient once population screening is instituted” [62]. Many examples support this notion, demonstrating that knowledge on rare diseases, which is commonly based on pre-NBS cohorts, is fostered by NBS, challenging relevant disease-specific aspects such as prevalence, phenotypic diversity, case definition, indications for treatment, disease variants, long-term outcome, response to treatment, and genotype–phenotype association [44,54,59]. In particular, NBS shifts the known phenotypic spectrum of these entities to attenuated or even benign phenotypes [44,62]. By reducing infantile mortality, NBS also improves the systematic study of severity-adjusted long-term outcomes [18,74,75,76,77,78,79].

Another stumbling block is the lack of specificity of a “recognizable latent or early symptomatic stage” (#4), since this period may range from days to years [62]. Furthermore, in disease groups like organic acidemias, whose latent period often ends during the newborn period, the opportunity for reliable pre-symptomatic identification is limited [80]. In line with this, a recent study demonstrated that 14.7% of individuals at risk actually experienced metabolic decompensation before NBS results were available [10].

There is also uncertainty about how “an important health problem” (#1) should be measured and quantified, resulting in ambiguity and different interpretations of disease severity at the level of the affected individual and at the population level [37,67].

#### 3.2.2. Diagnostic Aspects

Among the WJ criteria related to diagnostic aspects of the screening process (#3, #5, #6, #10; Figure 2), the definition of a “suitable test or examination” (#5), which “should be acceptable to the population” (#6), is considered vague and subjective. Suitability might relate to analytical and clinical validity in modern terminology. However, the definition of sufficient validity is still subjective and varies by test and condition [65]. Sensitivity and specificity can only be calculated with appropriate data [59] and might change with the start of NBS programs due to the concomitant broadening of the known phenotypic diversity. The sensitivity of NBS tests is generally high if cut-off values are set properly [44], but the major challenge is to improve specificity and positive prediction through reductions in false positives [44]. Furthermore, a suitable screening test always requires valid confirmatory diagnostic testing, which should ideally be conducted “with minimum delay, both to minimize stress to the parents and to allow treatment to proceed with greater confidence” [56]. Even in current NBS programs, the confirmation of a suspected disease can be difficult [56]. In future gNBS programs, the challenge of correct confirmation should not be under-estimated since, for many disorders, data on gene–disease association are vague and based on a small number of reported cases, the classification of individual gene variants is rather incomplete and sometimes incorrect, non-genetic biomarkers are unknown for a relevant number of diseases, and data on age-dependent penetrance and the expressivity of individual symptoms of genetic diseases are scarce.

The WJ criteria include the broad and imprecise statement that “facilities for diagnosis and treatment should be available” (#3). The failure to mention accessibility as an important factor has been pointed out by one study, whose authors stated that “access to experienced specialists varies and may require patients to travel if they are in rural areas” [44].

#### 3.2.3. Therapeutic Aspects

The treatability of a disease (including availability of and access to effective treatment) is a major prerequisite for the consideration of target diseases in current NBS programs [65]. The overall importance of treatability is reflected by three of the ten WJ criteria focusing on therapeutic aspects (#2, #3, #8; Figure 2). However, the term treatability itself requires specification, since available therapies do not always protect against the progression of a disease, but attenuate or delay the onset of a disease-specific phenotype [81].

Technological advances, such as gNBS, will enable the massive extension of NBS panels to include diseases which are not currently treatable but which are medically actionable in other ways [35,44]. Although the lack of access to or availability of effective therapies may cause parental anxiety and disappointment [44,59,82], parents may value the prevention of a traumatizing diagnostic odyssey [57,83,84] and the opportunity for improved family planning, even in the absence of effective therapies [35,48,54]. Furthermore, the value of identifying individuals with currently untreatable diseases who could be potentially recruited to innovative clinical trials is a delicate balance, and hence is discussed controversially.

Another crucial therapeutic challenge is the exact case definition of “whom to treat as a patient” (#8), or, in other words, to answer the question “What constitutes a positive case?” [54,55]. In addition, the early prediction of variant disease courses is important to individualize the start and intensity of treatment. In contrast to the traditional paradigm of medicine where symptoms guide diagnosis and treatment, NBS provides the challenging opportunity to identify individuals at risk of a target disease before the onset of symptoms [1]. Consequently, it is crucial that available tests reliably distinguish between healthy individuals and those requiring medical treatment. Since the association between genotype and clinical phenotype is often incompletely understood, and non-genetic diagnostic biomarkers are unavailable for many candidate target diseases of future gNBS programs, the use of gene variants as first tiers of identification in future NBS programs necessitates the establishment of reliable strategies for the confirmation of a suspected diagnosis [54,55].

#### 3.2.4. Economical Aspects

This sub-category is addressed by a single WJ criterion focusing on the cost of case-finding in relation to expenditure on medical care (#9; Figure 2). Some authors remark that not only NBS but any healthcare program is incomplete without the consideration of financial costs [48]. Economic evaluation is needed; however, decisions on new target diseases for NBS are complex, and hence should be based on multiple domains and should consider other aspects, such as medical, societal, and psychological costs (“cost of harm”). Furthermore, the subsequent costs of missed cases and false-positive NBS results should be evaluated in a full-cost model [44,48].

### 3.3. Missing Criteria and Sub-Categories

Studies on the WJ criteria have repeatedly emphasized their lack of completeness and, as a consequence, the literature on screening criteria produced a total of more than 50 different criteria lists and proposed close to 400 unique principles clustered in the abovementioned four major sub-categories. Missing criteria and sub-categories were also highlighted [33,34,36,40,48]. Among the missing criteria and sub-categories, a major gap was identified at the level of program management [33]. WJ criterion #10 states that “screening should be a continuing process and not a ‘once and for all’ project” (#10); however, it remains entirely unclear how to achieve this goal, i.e., how NBS programs should be developed, implemented, managed, evaluated, and continuously optimized in a balanced and transparent way [33,34,36]. Some studies highlighted the importance of (a) defining screening objectives and target populations of NBS before the start of screening; (b) developing a mechanism that enables the systematic collection of key data required for (c) the iterative and evidence-based evaluation and optimization of the quality, safety, and effectiveness of NBS programs; (d) ensuring that caregivers can make informed choices and promoting equity; (e) involving consumers and relevant stakeholders in screening policy-making; and (f) thus considering the ethical, legal and societal implications of NBS programs, to name but a few [5,33,34,35,36,44,54,61,63,65].

## 4. Discussion

### 4.1. Are We Moving towards Consensus on the Selection of Target Diseases for NBS Programs?

In their landmark publication, Wilson and Jungner defined a set of criteria that enabled the selection of conditions suitable for population-based mass screening based on the availability of suitable tests and acceptable treatment, the capacity of specialized centers, agreed case definitions and therapeutic decision-making, and balanced expenditures for case-finding and medical care [32]. They were very well aware of the need to find a balance between “in theory, screening is an admirable method” and “in practice, there are snags” [32]. In other words, although the idea of early disease detection and treatment seems essentially simple, “the path to its successful achievement […] is far from simple though sometimes it may appear deceptively easy.” [32]. Fifty-six years after their publication and subsequent discussions, with a plethora of proposed alternative screening lists [33,35,36,37,73], there is increasing discordance at the level of national policy-making despite theoretical agreement on the interpretation and actual use of the screening criteria. As a consequence, there is enormous variation in NBS programs worldwide [57,85,86,87,88,89]. Despite shared selection criteria, this discordance is also echoed by current gNBS pilot studies, which include a median number of close to 500 gene–disease combinations but can only agree on approximately 50 consensus gene–disease combinations. Noteworthily, the largest part of this consensus panel is made up of inherited metabolic diseases and other diseases already included in current NBS programs [22,90]. Therefore, it is high time to agree on a multi-dimensional framework for future NBS programs by evaluating and integrating previously proposed suggestions for a revision and extension of the WJ criteria, and overcoming reasons that may preclude a harmonized international approach to the development of NBS programs, such as a lack of precision and transparency, the exclusion of relevant stakeholders in policy-making, a lack of evidence of the quality, safety, and effectiveness of NBS, and the failure to place sufficient weight on the ethical, legal, and societal aspects of NBS [33,34,36].

### 4.2. In Practice, There Are Snags: The Importance of Closing the Knowledge Gap

The conceptual limitations of the WJ criteria unraveled by this systematic literature analysis are accompanied by actual limitations, highlighted by observational studies focusing on real-world evidence from NBS. These studies elucidate a major problem of all current NBS programs, i.e., the knowledge gap on the phenotypic diversity of rare diseases [78,91,92,93,94,95]. The WJ criteria were introduced at a time when the terms and concepts of “orphan disease”, “rare disease”, and “orphan drug” had not been introduced, and medicine had a clear focus on widespread diseases but virtually neglected rare diseases. Therefore, it is not unreasonable that Wilson and Jungner did not anticipate that applying their screening criteria to NBS programs for pediatric rare diseases could lead to significant problems or dilemmas. In rare diseases, clinical severity tends to be overestimated initially due to selection bias. With the implementation of NBS, the phenotypic spectrum of target diseases is commonly extended towards attenuated variants, with concomitant uncertainty about case definitions and indications for treatment (Figure 3). This uncertainty increases the risk of medicalization and over-treatment for individuals with attenuated disease variants, or even benign conditions not requiring treatment [78,91,92,93,94,95]. These examples highlight that natural history studies of target diseases in pre-NBS cohorts should be considered as incomplete, requiring re-evaluation after their introduction to NBS programs to adjust case definitions and indications for treatment (Figure 3). In addition to the structured evaluation of clinical long-term outcomes, the safety and effectiveness of treatments, and patient-reported outcomes of individuals identified by NBS, innovative approaches are required to improve diagnostic quality, confirmation, case definition, and the early prediction of inherent disease severity in asymptomatic newborns with a suspected diagnosis of a target disease. Recommendations for the confirmation of a suspected diagnosis and uniform case definitions have been introduced [96,97], and various technical innovations have been developed to achieve this goal, such as second- and multiple-tier strategies [6,9,98,99,100]. Other innovative strategies are currently under investigation, such as digital tiers using machine learning methods [101,102,103], and combined prediction models for clinical severity [74,75,76,77,78]. Noteworthily, the development of reliable case definitions and strategies for confirmatory diagnosis in advance of screening will also be key to the success of future gNBS programs, and hence should be considered essential for the selection of new target diseases.

### 4.3. Managing Change as a Project: NBS as an Integrated Public Health Program

Wilson and Jungner highlighted that “Case finding should be a continuing process and not a ‘once and for all’ project” (#10; [32]). In other words, NBS as a whole should be developed and organized as a learning system that enables continuous data-driven evaluation and optimization. To achieve this goal, a recent concept paper proposes to organize NBS as an integrated public health program based on central coordination with a standardized core structure; data-driven evaluations of diagnostic quality, safety, and (cost-)effectiveness; and continuous quality management [104]. This notion is supported by previous clinical studies in NBS cohorts demonstrating that real-world evidence from structured longitudinal data collection is a valuable source of evidence to guide the development and iterative optimization of NBS strategies [1]. It has been shown that systematic data collection and longitudinal follow-up help to improve process quality [10]; progress the understanding of natural history, case definition, and phenotype prediction in individuals with target diseases identified by NBS [74,75,76,77,78,93]; elucidate the impact of therapeutic quality on long-term outcomes in NBS cohorts [10,12,13,18]; aid in the development of evidence-based guidelines [105]; enable cost-effectiveness analysis in NBS cohorts [106]; stimulate research [107]; and allow researchers to learn about parental and societal expectations of and perspectives on NBS [108,109,110,111]. Therefore, the major advantage of organizing NBS as an integrated public health program is the establishment of a learning system that helps to close the knowledge gap on NBS target diseases and to manage change, as well as optimize the quality, safety, and (cost-)effectiveness of NBS programs based on real-world evidence.

### 4.4. Limitations

Although the study was conducted as a systematic review, the study design is based on qualitative text analysis, and thus the study comprises a collection of comments and critiques on WJ criteria and their applicability in current and future NBS programs. Thus, no metric parameters were available, and no statistical analysis or meta-analysis was possible to enhance evidence.

## 5. Conclusions: Developing a Multi-Dimensional Framework for Future NBS Programs

Many proposals for adapting the WJ criteria to our time, in order to reflect topics that have emerged in medicine since the publication of the WJ criteria back in 1968, have been presented [32]. These proposals touch upon evidence-based medicine and healthcare, informed choice, the involvement of relevant stakeholders and the public in policy-making, cost-effectiveness, and quality assurance, to name but a few [34,35,36]. Any revision of screening criteria should refer to these modern requirements within a multi-dimensional framework, integrating not only the medical perspective, but also ethical, legal, and societal perspectives, and should give attention to patients, relevant stakeholders, and the public. This could be the path to make sure that the benefits of NBS programs outweigh the harms, and to secure the high public acceptability of this successful program of secondary prevention in the future. Using this approach, the authors—in collaboration with patient organizations and other stakeholders—have formed the NEW_LIVES project group, and are currently developing a revised set of screening criteria.

## Figures and Tables

**Figure 1 IJNS-10-00062-f001:**
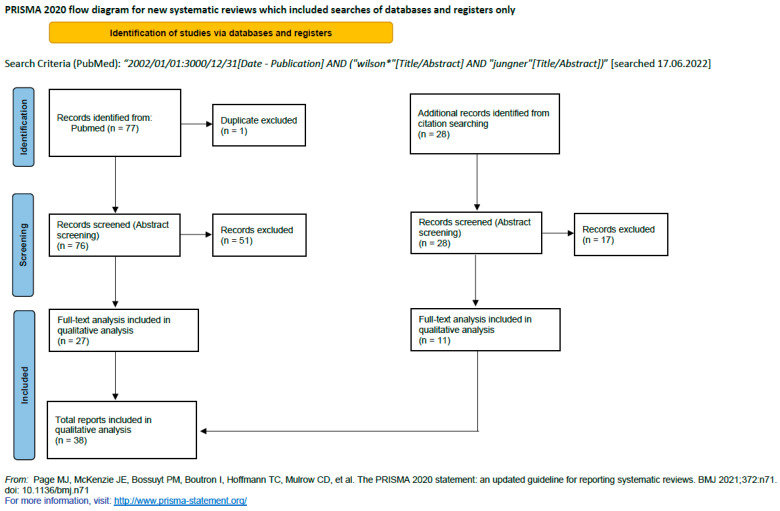
PRISMA diagram of the literature search [38].

**Figure 2 IJNS-10-00062-f002:**
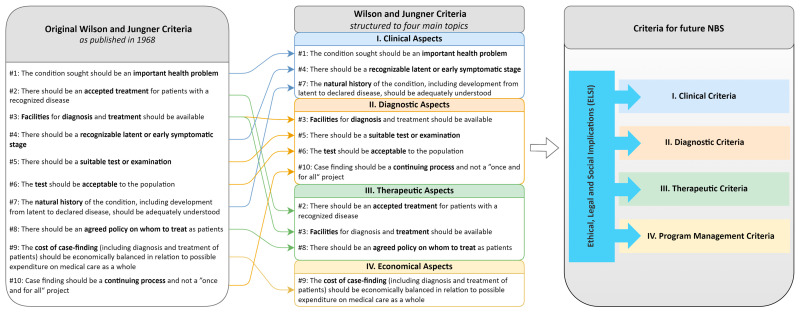
Revising the screening criteria. *Principles and Practice of Screening for Disease*, as published by Wilson and Jungner in 1968 [32], includes 10 specific criteria which have been the basis of all NBS programs worldwide for more than 50 years (**left**). In subsequent studies, these criteria have been commonly re-arranged in four different sub-categories (**middle**). The systematic literature review not only identified relevant shortcomings of single WJ criteria, but also highlighted their missing focus on complex aspects of program management and the insufficient systematic consideration of ethical, legal, and societal implications (ELSI), which should form the basis of all NBS programs (**right**). A multi-dimensional framework integrating all relevant perspectives would be an excellent opportunity to revise the original WJ criteria and to make them fit for the demands and further developments of NBS programs. Figure was created with draw.io (https://drawio-app.com/, accessed on 17 July 2024).

**Figure 3 IJNS-10-00062-f003:**
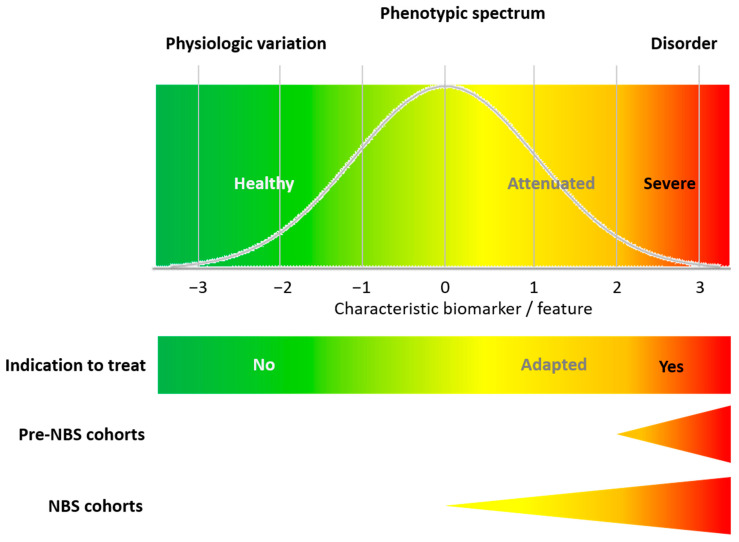
Extension of the phenotypic spectrum of an NBS target disease after NBS implementation. Introduction to NBS expands the phenotypic spectrum towards attenuated disease variants. While severe and attenuated variants are usually easy to distinguish, the exact differentiation between healthy and attenuated forms can be challenging, requiring risk-stratified and individualized treatment indications.

## Data Availability

No new data were created in this study. Data sharing is not applicable to this article.

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
