# Peer review of "Wilson and Jungner Revisited: Are Screening Criteria Fit for the 21st Century?"

_2409-515X, 2024, doi:10.3390/ijns10030062_

Round 1
Reviewer 1 Report
Comments and Suggestions for Authors
This manuscript provides a summary of aspects of the Wilson-Jungner criteria in the era of NGS. The manuscript reads well but in my view, the conclusion is quite vague and it would be strengthened by a more in depth discussion on what NGS brings to then field and how this new technology will be integrated into newborn screening, both in terms of what diseases to screen for (we essentially do not know as yet but time will tell) and ethical aspects !
Author Response
Reviewer 1: This manuscript provides a summary of aspects of the Wilson-Jungner criteria in the era of NGS. The manuscript reads well but in my view, the conclusion is quite vague and it would be strengthened by a more in depth discussion on what NGS brings to then field and how this new technology will be integrated into newborn screening, both in terms of what diseases to screen for (we essentially do not know as yet but time will tell) and ethical aspects !
Response: We thank you for your overall positive comments. The manuscript reports the results of the first part of an ongoing collaborative study focusing on the overarching topic of evaluating, revising and applying screening criteria. We are very well aware that because of this procedure some aspects and conclusions are more theoretically discussed in the first part (i.e., the submitted manuscript). However, the major aim of the first part is to systematically evaluate major limitations and gaps of the original Wilson & Jungner criteria based on published literature and to promote the process of revising them. However, the revised set of criteria, which is currently developed within a multi-dimensional framework, will be published separately (second part). In the third part, we will apply these criteria to propose a transparent future extension list of gene/disease pairs and to evaluate previously reported lists from gNBS pilot studies.
The manuscript should not be read as particularly breaking a lance for introducing NGS to NBS programs. NBS has been challenged by new opportunities since the start of these programs, necessitating careful evaluation of potential candidates for an extension of existing programs. Regardless of NGS, we would like to stress the point (like other authors did before) that screening criteria and the level of development of the regulatory framework guiding NBS programs cannot be regarded as fit for any substantial extension of NBS programs at all. Therefore, we would like to stress the point that a revision of the regulatory basis of NBS programs is overdue.
Reviewer 2 Report
Comments and Suggestions for Authors
Many thanks to bring this issue to the table. I really have enjoyed reading your statement, and value its well timed relevance. Being involved in a ‘prioritisation’ board for NBS, i fully recognize the thin line between what we can do, how we can improve outcome, as well as how screening also results in a number of unintended ‘side’ effects.
I agree that a major obstacle in the incomplete understanding of the natural history and phenotypic diversity of rare diseases, but that is not limited to molecular genetics, as e.g. MCAD screening had resulted in a lot of ‘new’ previously not recognized cases, resulting in both a reduction in 'sudden death’ as well as much more cases with more ‘preventive’ hospitalisations’.
A more classic ‘genomic’ example is SMA screening, resulting in incomplete (sensitivity) as well in overdiagnosis (‘specificity’). For both examples, the benefits are rather clear at the population level, while this is occasionally less clear at the individual family level.
I somewhat disagree with the statement in the abstract that technological innovations are the basis of expanding. In my assessment, the statement in full version is more accurate, ‘driven by’ or ‘supported by’ as insights in the mechanisms of the disease are in my assessment more the basis, the technics the tools.
Besides the concept of ‘to minimize their risk of harm’, there is also the risk of reduced adherence to the NBS.
The diagnostic delay is sometimes used as an argument, but a diagnosis does not necessary result in effective, available and accessible treatment, and ‘detection of cases for potential clinical trials’ is a very delicate balance, somewhat advocated by EURORDIS.
Related to this, I highly recommend to somewhat better explain Figure 1. In its current version, is somewhat suggest a future application of the criteria, but it is not yet clear to this reviewer how this has been reconstructed. The legend has one reference (32, going back to 1968).Perhaps you should consider repositioning this figure in the discussion section of the paper ?
We do need a more detailed description of the systematic review, or otherwise, this should be addressed as a structured search (two pair of eyes, sources used, quality assessment, agreement). This would still remain a very useful exercise, but accuracy in the use of ‘systematic review’ is warranted. At present, it does not yet read as a systematic review in its methods.
Line 136: i understood that galactosemia has a similar pattern.
On therapeutic aspects, this also covers availability and access (affortability) in my reading. I agree that (some, likely the majority) may value, but it is not impossible that a relevant portion does not value an asymptomatic diagnosis (even latent up to adulthood).
Another ‘snag’ is the uncertainties related to ‘findings by GWS’ and phenotypic range.
This paper obviously does not ‘solve’ the issue, but at least bring the topic to the forefront in the relevant journal to this field.
Author Response
Comment 1: Many thanks to bring this issue to the table. I really have enjoyed reading your statement, and value its well timed relevance. Being involved in a ‘prioritisation’ board for NBS, i fully recognize the thin line between what we can do, how we can improve outcome, as well as how screening also results in a number of unintended ‘side’ effects.
I agree that a major obstacle in the incomplete understanding of the natural history and phenotypic diversity of rare diseases, but that is not limited to molecular genetics, as e.g. MCAD screening had resulted in a lot of ‘new’ previously not recognized cases, resulting in both a reduction in 'sudden death’ as well as much more cases with more ‘preventive’ hospitalisations’.
A more classic ‘genomic’ example is SMA screening, resulting in incomplete (sensitivity) as well in overdiagnosis (‘specificity’). For both examples, the benefits are rather clear at the population level, while this is occasionally less clear at the individual family level.
Response 1: We thank you for your overall positive comments and the important thoughts on the subject.
Comment 2: I somewhat disagree with the statement in the abstract that technological innovations are the basis of expanding. In my assessment, the statement in full version is more accurate, ‘driven by’ or ‘supported by’ as insights in the mechanisms of the disease are in my assessment more the basis, the technics the tools.
Response 2: We agree with your comment. Technological innovations should not be understood as an end in itself but as an opportunity. We have changed the sentenced accordingly.
Comment3: Besides the concept of ‘to minimize their risk of harm’, there is also the risk of reduced adherence to the NBS.
Response 3: We fully agree with your concern. Therefore, we have included the following sentence in the conclusion: “This could be the path to make sure that benefits of NBS programs outweigh harms and to secure the high public acceptability of this successful program of secondary prevention in the future.“
Comment 4: The diagnostic delay is sometimes used as an argument, but a diagnosis does not necessary result in effective, available and accessible treatment, and ‘detection of cases for potential clinical trials’ is a very delicate balance, somewhat advocated by EURORDIS.
Response 4: Excellent point! We have included a short statement on this in paragraph 3 of the Results (Therapeutic aspects): “Furthermore, the value of identifying individuals with currently untreatable diseases who could be potentially recruited to innovative clinical trials is a delicate balance and hence is discussed controversially.”
Comment 5: Related to this, I highly recommend to somewhat better explain Figure 1. In its current version, is somewhat suggest a future application of the criteria, but it is not yet clear to this reviewer how this has been reconstructed. The legend has one reference (32, going back to 1968). Perhaps you should consider repositioning this figure in the discussion section of the paper ?
Response 5: We adapted the figure and the figure legend to allow a better understanding and shifted the figure within the manuscript towards results and discussion (We left the formatting of the figure position for the editorial team). This manuscript should be regarded as the results of the first part of a study focusing on the evaluation, revision and application of screening criteria. The first part (this manuscript) focuses on a literature-based evaluation of limitations and gaps of screening criteria and to promote a procedure of revising them. An updated set of criteria is currently developed within a multi-dimensional framework and will be published separately (second part).
Comment 6: We do need a more detailed description of the systematic review, or otherwise, this should be addressed as a structured search (two pair of eyes, sources used, quality assessment, agreement). This would still remain a very useful exercise, but accuracy in the use of ‘systematic review’ is warranted. At present, it does not yet read as a systematic review in its methods.
Response6: Following the PRISMA 2020 guideline we are convinced the study to be a systematic review and added all missing aspects required by the PRISMA 2020 guideline (Flow diagram and checklist) and adapted the Abstract, Introduction, Methods, Results and Discussion and Statements accordingly.
Comment 7: Line 136: i understood that galactosemia has a similar pattern.
Response 7: Unfortunately, we did not get the point of this comment and, therefore, did not changed this sentence.
Comment 8: On therapeutic aspects, this also covers availability and access (affortability) in my reading. I agree that (some, likely the majority) may value, but it is not impossible that a relevant portion does not value an asymptomatic diagnosis (even latent up to adulthood).
Response 8: We thank you for this important aspect of availability and access of therapies and included this in the paragraph “Therapeutic aspects” in the Results.
Comment 9: Another ‘snag’ is the uncertainties related to ‘findings by GWS’ and phenotypic range.
Response 9: This point has already been addressed in paragraph 2 of the Results (Diagnostic aspects): “In future gNBS programs, the challenge of correct confirmation should not be under-estimated since for many disorders data on the gene-disease association are vague and based on a small number of reported cases, the classification of individual gene variants is rather incomplete and sometimes incorrect, non-genetic biomarkers are unknown for a relevant number of diseases, and data on age-dependent penetrance and expressivity of individual symptoms of genetic diseases is scarce.”
Comment 10: This paper obviously does not ‘solve’ the issue, but at least bring the topic to the forefront in the relevant journal to this field.
Respponse 10: We thank the reviewer for this comment.
Round 2
Reviewer 2 Report
Comments and Suggestions for Authors
the authors have addressed my suggestions, nothing to add